# Improved transcriptome assembly using a hybrid of long and short reads with StringTie

**Alaina Shumate**[1,2], **Brandon Wong**[1,2,3,4], **Geo Pertea**[5], **Mihaela Pertea**[1,2,3]*

**1** Department of Biomedical Engineering, Johns Hopkins University, Baltimore, Maryland, United States of America, **2** Center for Computational Biology, Johns Hopkins University, Baltimore, Maryland, United States of America, **3** Department of Computer Science, Johns Hopkins University, Baltimore, Maryland, United States of America, **4** Department of Applied Math and Statistics, Johns Hopkins University, Baltimore, Maryland, United States of America, **5** The Lieber Institute for Brain Development, Baltimore, Maryland, United States of America

* mpertea@jhu.edu

## Abstract

Short-read RNA sequencing and long-read RNA sequencing each have their strengths and weaknesses for transcriptome assembly. While short reads are highly accurate, they are rarely able to span multiple exons. Long-read technology can capture full-length transcripts, but its relatively high error rate often leads to mis-identified splice sites. Here we present a new release of StringTie that performs hybrid-read assembly. By taking advantage of the strengths of both long and short reads, hybrid-read assembly with StringTie is more accurate than long-read only or short-read only assembly, and on some datasets it can more than double the number of correctly assembled transcripts, while obtaining substantially higher precision than the long-read data assembly alone. Here we demonstrate the improved accuracy on simulated data and real data from *Arabidopsis thaliana*, *Mus musculus*, and human. We also show that hybrid-read assembly is more accurate than correcting long reads prior to assembly while also being substantially faster. StringTie is freely available as open source software at https://github.com/gpertea/stringtie.

**Data Availability Statement:** StringTie is freely available as open source software at https://github.com/gpertea/stringtie.

**Funding:** This study was funded by the National Science Foundation (grant DBI-1759518) awarded

## Author summary

Identifying the genes that are active in a cell is a critical step in studying cell development, disease, the response to infection, the effects of mutations, and much more. During the last decade, high-throughput RNA-sequencing data have proven essential in characterizing the set of genes expressed in different cell types and conditions, which has driven a strong need for highly efficient, scalable and accurate computational methods to process these data. As sequencing costs have dropped, ever-larger experiments have been designed, often capturing hundreds of millions or even billions of reads in a single study. These enormous data sets require highly efficient and accurate computational methods for analysis, and they also present opportunities for discovery. Recently developed long-read technology now allows researchers to capture entire transcripts in a single long read, enabling more accurate reconstruction of the full exon-intron structure of genes, although these reads have higher error rates and higher costs. In this study we use the high accuracy

to MP. The funders had no role in study design, data collection and analysis, decision to publish, or preparation of the manuscript.

**Competing interests:** The authors have declared that no competing interests exist.

of short reads to correct the alignments of long RNA reads, with the goal of improving the identification of novel gene isoforms, and ultimately our understanding of transcriptome complexity.

This is a *PLOS Computational Biology* Software paper.

## Introduction

Uncovering the transcriptome of an organism is crucial to understanding the functional elements of the genome. This requires being able to accurately identify transcript structure and quantify transcript expression levels. In eukaryotes, this task is more challenging due to alternative splicing. It occurs frequently, with an estimated 92%-94% of human genes undergoing alternative splicing [1]. Short-read RNA-sequencing (RNA-seq) has been a useful tool in uncovering the transcriptome of many organisms when coupled with computational methods for transcriptome assembly and abundance estimation. Short-read sequencing provides the advantage of deep coverage and highly accurate reads. Second-generation sequencers such as those from Illumina can produce millions of reads with an error rate of less than 1% [2]. While second-generation sequencers produce very large numbers of reads, their read lengths are typically quite short, in the range of 75–125 bp for most RNA-seq experiments today. These short reads often align to more than one location in the genome, and also suffer the limitation that they rarely span more than two exons, resulting in a difficult and sometimes impossible task of constructing an accurate assembly of genes with multiple exons and many diverse isoforms, no matter how deeply those genes are sequenced. These issues can be alleviated by third-generation sequencing technologies such as those from Pacific Biosciences (PacBio) and Oxford Nanopore Technologies (ONT). Reads from these technologies can be greater than 10 kilobases long, allowing full-length transcripts to be sequenced. However, practical limitations often impede the ability to capture full-length transcripts. These include the rapid rate of RNA degradation, shearing of the RNA during library preparation, or incomplete synthesis of cDNA [3]. Additionally, long reads have a high error rate relative to Illumina short reads [4], and the throughput of long-read RNA-seq is much lower than that of short-read RNA-seq. This can make it difficult in some cases to define precise splice sites. Using a combination of short reads and long reads for transcriptome assembly allows us to take advantage of the strengths of each technology and mitigate the weaknesses. While there are many tools that use either short reads or long reads for transcriptome assembly and quantification, there are very few that use a hybrid of the two. These tools include Trinity [5], IDP-denovo [6], and rnaS-PAdes [7], which only perform *de novo* transcriptome assembly. If a high-quality reference genome of the target organism is available, as it is for human and for a large number of plants, animals, and other species, *de novo* transcriptome assembly usually produces lower-quality assemblies compared to reference-based approaches. This is due to technical challenges resulting from the presence of gene families, large variations in gene expression, and extensive alternative splicing [8]. StringTie is a reference-based transcriptome assembler that can assemble either long reads or short reads, and has been shown to be more accurate than existing short and long read assemblers [9].

In this work we present a new release of StringTie which allows transcriptome assembly and quantification using a hybrid dataset containing both short and long reads. We show with

simulated data from the human transcriptome that hybrid-read assemblies result in more accurate assembly and coverage estimates than using long reads or short reads alone. Additionally, we evaluate the assembly accuracy on 9 real datasets from 3 well-studied species (human, *Mus musculus*, and *Arabidopsis thaliana*) and demonstrate that the hybrid-read assemblies are more accurate than both the long-read only and short-read only assemblies. We also demonstrate that hybrid-read assembly is more accurate and also substantially faster than a strategy of correcting long reads prior to assembly.

## Results

Our hybrid transcriptome assembly algorithm takes advantage of the strengths of both long and short read RNA sequencing, by combining the capacity of long reads to capture longer portions of transcripts with the high accuracy and coverage of short-read data to produce better transcript structures as well as better expression estimates. Fig 1A shows examples of alignment artifacts that are often present in long reads because of the high error rate. These include "fuzzy" splice sites as well as retained introns, spurious extra exons, falsely skipped exons, and false alternative splice sites. Fig 1B shows a specific example of a 9-exon isoform of a human gene that can only be correctly assembled using both long and short reads. There are no long reads mapped to the first 3 exons of this isoform, and we see a retained intron in the alignment. Among the short-read alignments, the 4th and 7th introns are only spanned by a single spliced read, and exons 5 and 8 are not completely covered. This causes the transcripts to be assembled in 3 fragments. Using both long and short reads we were able to correctly assemble the transcript by using the short reads to support the splice sites found in the long-read alignments (See Methods). The adequate short-read coverage of exons 1–3 also allowed us to assemble these despite the lack of coverage in the long reads.

Next, we present results for StringTie's performance with hybrid long and short read sequences on simulated data as well as on three real RNA-seq data sets, from human, mouse, and the model plant *Arabidopsis thaliana*.

### Simulated data

Since it is not possible to know the true transcripts that are present in real RNA-seq datasets, we first used simulated data to assess the accuracy of hybrid-read assembly and quantification across the transcriptome. To this end, we simulated two human RNA-seq datasets, one with short-reads and one with ONT direct RNA long reads (see Methods) and assembled them with StringTie.

To evaluate the accuracy of hybrid-read assemblies compared to long-read only and short-read only assemblies, we generated 4 different assemblies of each read type (long, short, and hybrid) with 4 different sets of parameters (Fig 2A). We then computed the precision and sensitivity for each assembly. Precision is defined as the percent of assembled transcripts that match a true transcript, and sensitivity is defined as the percent of true transcripts that match an assembled transcript (see Methods). For these calculations, we considered a transcript to be truly expressed only if it was fully covered by either the short or long simulated reads. For each hybrid-read assembly, we calculated the relative percent increase in precision and sensitivity over the long-read and short-read assemblies with the same parameters (see Methods). When we report the percent increase of any metric, we are referring to the *relative* percent increase. Averaging these results, we saw that hybrid-read assemblies had an increase in precision of 9.8% over the long-read assemblies, and an increase in sensitivity of 24.4%. As compared to the short-read assemblies, the hybrid-read assemblies had an increase in precision of 12.5% and an increase in sensitivity of 22.1%. To confirm that these improvements were not simply

**A**

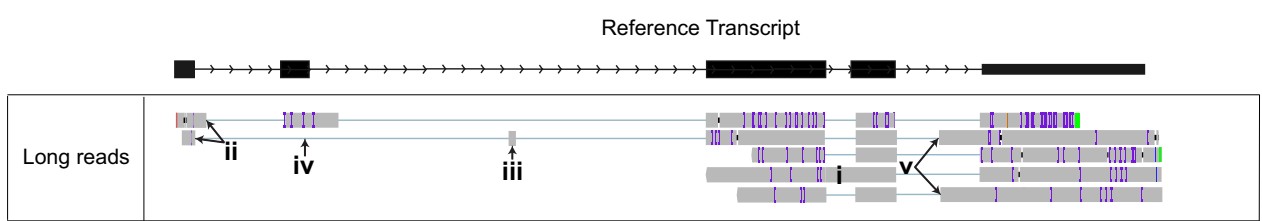

**B**

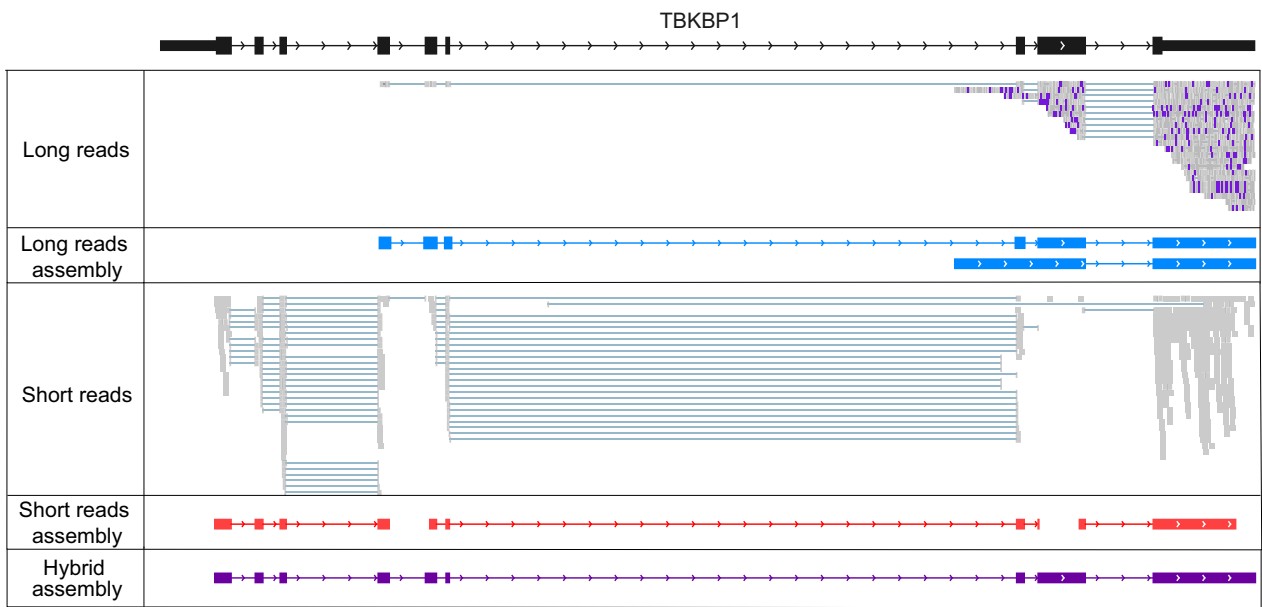

**Fig 1. A)** Artifacts present in the long read alignments: i) retained introns; ii) disagreement around the splice sites; iii) spurious extra exons; iv) falsely skipped exons; v) false alternative splice sites. **B)** Example of a human transcript that can only be correctly assembled using both the long and short reads. This is human transcript ENST000000361722.7 from the TBKBP1 gene. Blue lines in the middle of the reads (gray boxes) indicate a spliced alignment. Purple lines within the reads indicate mismatches in the alignment. The long reads alignments do not have coverage of exons 1–3 and contain a retained intron. The short-read alignments lack adequate splice-site support across the 4th intron and the 7th intron and do not have complete coverage of exons 5 and 8.

due to increased coverage in the hybrid reads, we performed the same experiment where the long, short, and hybrid-read datasets all had approximately equal coverage (See Methods and S1 File). When controlled for coverage, we still see that the hybrid-read assembly clearly outperforms both the long and short-read only assemblies in precision and sensitivity (S1 Fig).

We also compared the coverage computed by StringTie to the actual coverage of long-read only, short-read only, and hybrid-read assemblies created with default parameters (Fig 2B). StringTie's computed coverage of the hybrid-read assembly was closest to the true coverage. We found that the correlation between true and calculated coverage for hybrid-read assembly

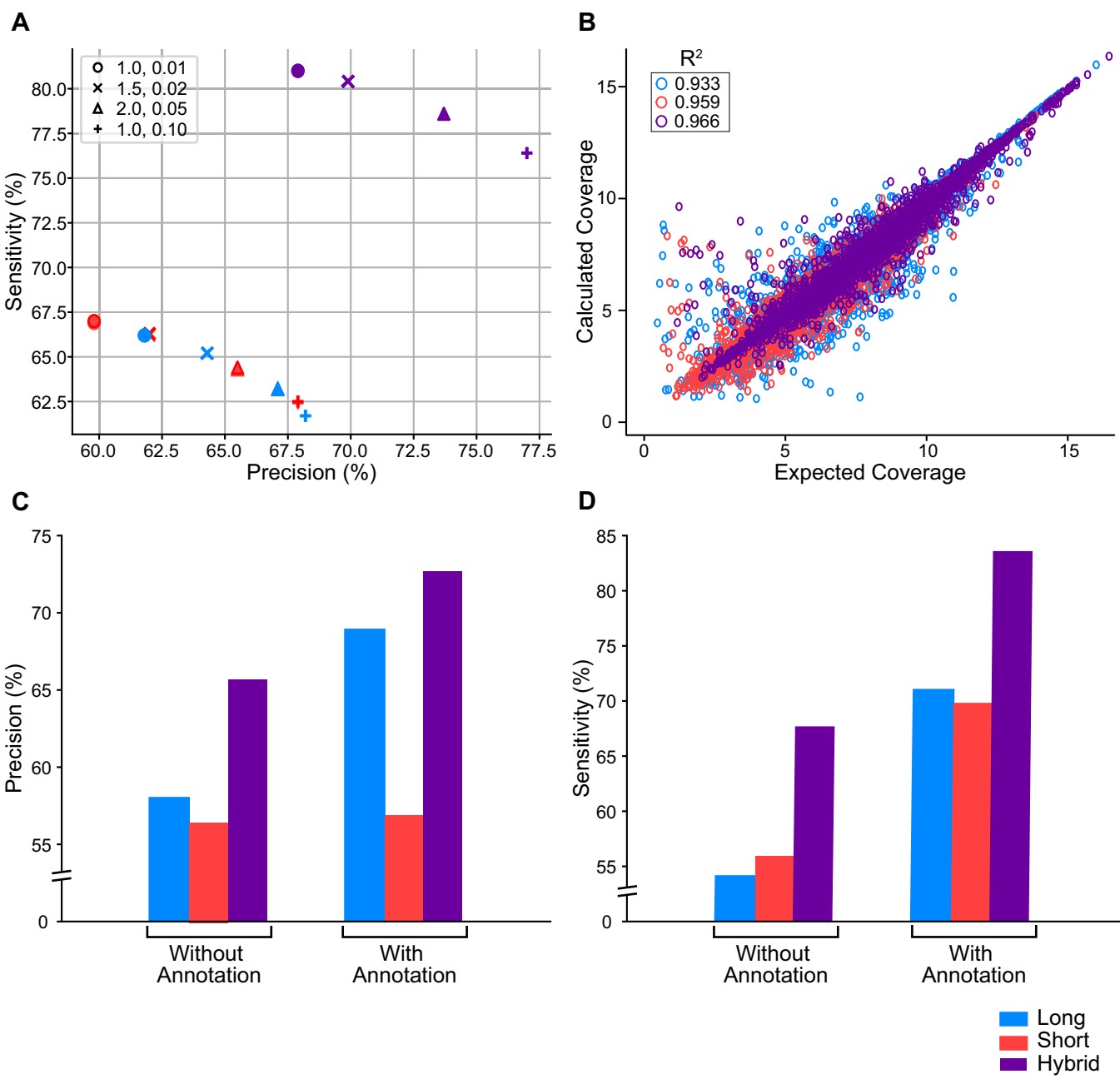

**Fig 2. A)** Sensitivity and precision for StringTie assemblies of simulated data with varying sensitivity parameters. The two StringTie parameters varied were the minimum read coverage allowed for a transcript (-c) and the minimum isoform abundance as a fraction of the most abundant transcript at a given locus (-f). Each shape represents a different combination of -c,-f parameters with the values indicated in the legend. The default values of -c and -f are 1.0 and 0.01 respectively and are represented by the circle marker. **B)** Calculated coverage vs. expected coverage for long-read, short-read, and hybrid-read assemblies of simulated data. Coverage values are normalized to $\log_2(1 + \text{coverage})$. **C)** Precision of long-read, short-read, and hybrid-read assemblies of simulated data with and without guide annotation. **D)** Sensitivity of long-read, short-read, and hybrid-read assemblies of simulated data with and without guide annotation.

yielded an $R^2$ value of 0.966, higher than the $R^2$ values for both the short-read (0.959) and long-read (0.933) only assemblies.

If the reference genome annotation is reliable, some methods (including StringTie) can use that annotation to improve the accuracy of the transcriptome assembly. Note that not all transcripts in the reference annotation will be expressed in the data, therefore the assembler needs

to accurately determine which of the transcripts are present in the data. Moreover, reference annotations are usually incomplete, so StringTie's default behavior when annotation is provided is to assume that novel transcripts could be present as well. We wanted to assess if String-Tie's performance improves on hybrid data if the human reference annotation is provided. As shown in Fig 2C and 2D, both precision and sensitivity improved when the reference annotation was provided, and hybrid data assembly had the highest sensitivity and precision regardless of whether the reference annotation was provided or not. The improvement in precision was insignificant with short-read data due to the fact that more annotated isoforms were assembled even though they were not expressed. This wasn't a problem with the other data sets, as long reads were able to better recover the full exon-intron structure of the expressed transcripts. The use of hybrid-read data plus annotation had an increase in precision of 10.7% and an increase in sensitivity of 23.5% as compared to the hybrid-read data assembled without annotation.

## Real data

Next, we evaluated the accuracy of hybrid-read assemblies on real data, which is in general much more challenging than simulated data, in part because the real data may contain biases or other artifacts not always captured by simulated data. From publicly available data, we chose a total of 9 combinations of long and short reads from 3 well-studied species: *Arabidopsis thaliana*, *Mus musculus*, and human. The Mus *musculus* samples include samples from brain and liver tissue. The human samples are from the NA12878 cell line. Each combination of long and short reads is derived from the same sample. All three species have well-characterized reference annotation available, even though their level of completeness is not fully established [10]. The short-read libraries were all generated through poly-A selection and sequenced with Illumina sequencers. The long reads were generated by a variety of technologies including ONT direct RNA, ONT cDNA, and PacBio cDNA (Table 1). The quality of these long reads varies with error rates ranging from 3.2% to 17.2% (S1a Table) and the percentage of reads that cover full-length isoforms ranges from 25.4% to 67.2% (S2 Table).

Although we cannot know exactly which transcript molecules are present in the samples, it can be assumed that an assembly with more transcripts matching known annotations is more sensitive, and an assembly is more precise if known transcripts comprise a higher percentage of the total number of assembled transcripts. Therefore, to evaluate the accuracy of the assemblies of real data, we report two values: (1) the number of assembled transcripts matching an annotated transcript, and (2) precision, which we define as the percentage of assembled transcripts matching known annotations. We chose to report the number of transcripts matching the annotation instead of sensitivity, because it is impossible to know exactly which transcripts are truly expressed in real experimental data. As with the simulated data, we report the relative percent increase/decrease of both metrics. Since short-read data offers much higher coverage of the expressed transcriptome, for these calculations we only consider loci with long-read coverage, but the results are similar when we look at all loci (S2 Fig).

We also compare hybrid-read assembly to the strategy of correcting long reads prior to assembly, which is a common approach to handling the high error rate of long reads. Multiple previous algorithms have been proposed to combine long and short reads into high-accuracy long reads [11], but those approaches were primarily intended to be applied to whole-genome data with the aim to improve the quality of genome assemblies. Only recently a new method, called TALC [12], was developed for long-read correction in the context of RNA-seq data by incorporating coverage analysis throughout the correction process. Using the corresponding short-read sample, we corrected each long-read sample with TALC. On average TALC

**Table 1. Availability of real RNA-seq datasets and descriptions of sequencing technology used including chemistry and basecaller version for ONT datasets.**

| Accession Number | Database | Species (Tissue) | Sequencing Technology |
|---|---|---|---|
| ERR3486096 | European Nucleotide Archive | *A. thaliana* | llumina HiSeq 4000 |
| ERR3764345 | European Nucleotide Archive | *A. thaliana* | ONT direct RNA SQK-RNA001 MinION Guppy v2.3.1 |
| ERR3486098 | European Nucleotide Archive | *A. thaliana* | llumina HiSeq 4000 |
| ERR3764349 | European Nucleotide Archive | *A. thaliana* | ONT direct RNA SQK-RNA001 MinION Guppy v2.3.1 |
| ERR3486099 | European Nucleotide Archive | *A. thaliana* | llumina HiSeq 4000 |
| ERR3764351 | European Nucleotide Archive | *A. thaliana* | ONT direct RNA SQK-RNA001 MinION Guppy v2.3.1 |
| ERR2680378 | European Nucleotide Archive | *M. musculus* (brain) | llumina HiSeq 4000 |
| ERR2680375 | European Nucleotide Archive | *M. musculus* (brain) | ONT direct RNA SQK-RNA001 MinION Albacore 2.1.10 |
| ERR2680377 | European Nucleotide Archive | *M. musculus* (brain) | ONT cDNA MinION SQK-PCS108 Albacore 2.1.10 |
| ERR2680380 | European Nucleotide Archive | *M. musculus* (liver) | Illumina HiSeq 4000 |
| ERR2680379 | European Nucleotide Archive | *M. musculus* (liver) | ONT direct RNA SQK-RNA001 MinION Albacore 2.1.10 |
| SRR4235527 | Sequence Read Archive | *H. sapiens* | Illumina Genome Analyzer IIx |
| NA12878-dRNA | github.com/nanopore-wgs-consortium | *H. sapiens* | ONT direct RNA SQK-RNA001 MinION Guppy v3.2.6 |
| NA12878-cDNA | github.com/nanopore-wgs-consortium | *H. sapiens* | ONT cDNA SQK-PCS108 MinION Albacore 2.1 |
| SRR1153470 | Sequence Read Archive | *H. sapiens* | llumina HiSeq 2000 |
| SRR1163655 | Sequence Read Archive | *H. sapiens* | PacBio cDNA PacBio RS |

decreased the error rate by 9.5% (S1b Table). We created additional long-read and hybrid-read assemblies with the TALC-corrected reads and then compared the accuracy of the hybrid-read assemblies to the corrected long-read assemblies. We also assessed whether using corrected long reads in a hybrid-read assembly substantially improved the accuracy. As we show below, TALC is quite effective at correcting errors; however, it is far slower than StringTie (running on a single RNA-seq samples takes TALC a day or longer, compared to less than one hour for StringTie), and it does not improve transcript assembly as compared to our new hybrid assembly algorithm.

## Arabidopsis thaliana

The hybrid-read assemblies of the *Arabidopsis thaliana* samples achieved higher precision and contained more annotated transcripts than both the long-read and short-read assemblies (Fig 3A–3C). The average percent increase in precision in the hybrid-read assemblies was 8.0%

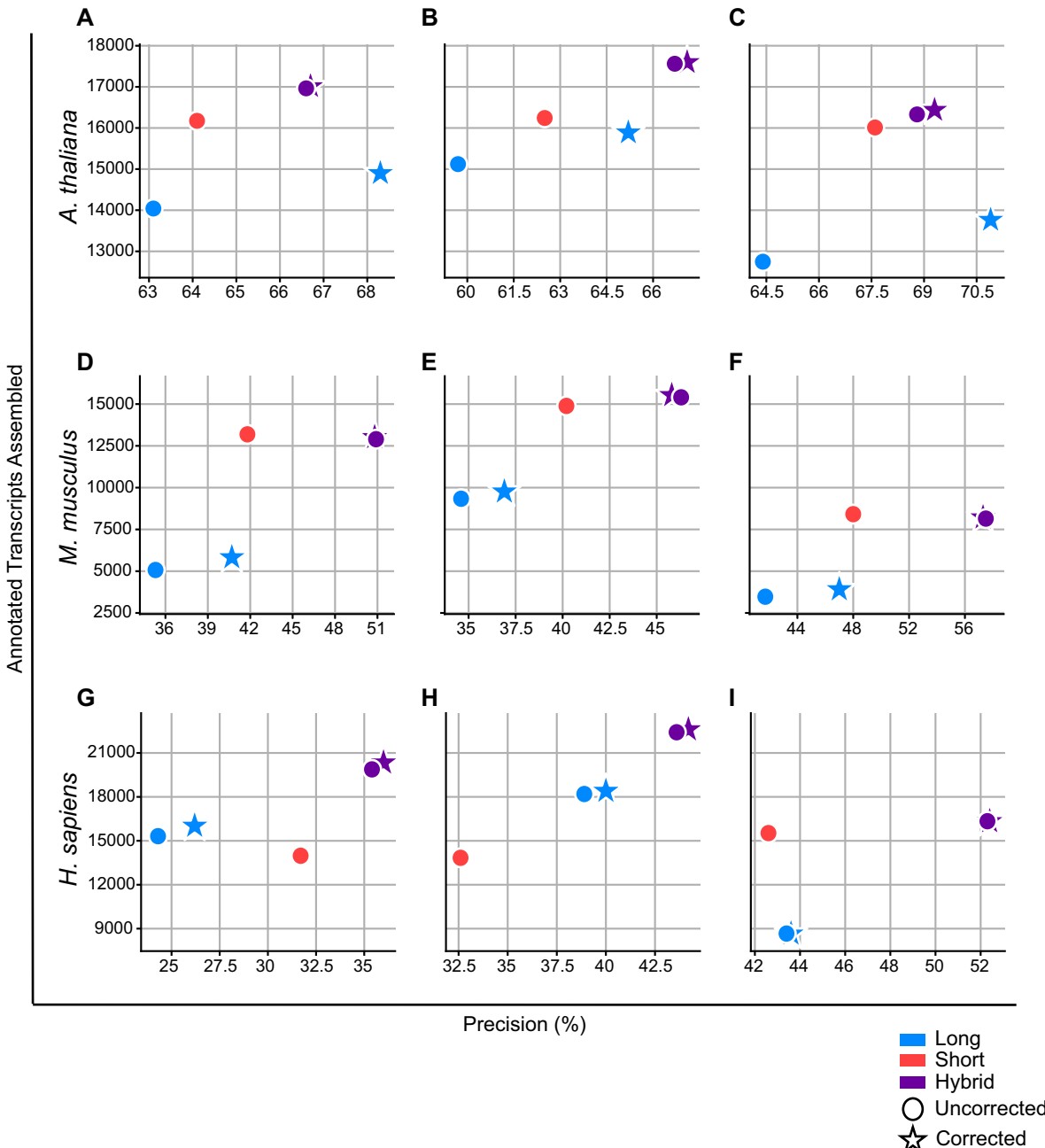

**Fig 3. Precision and the number of annotated transcripts assembled for 9 real datasets from *Arabidopsis thaliana*, *Mus musculus*, and human.** Only loci with long read expression are considered for these calculations The circle markers represent assemblies created from uncorrected reads, and the stars represent assemblies created from long-reads corrected with TALC. The long and short read combinations analyzed from *Arabidopsis thaliana* were **A)** ERR3486096 and ERR3764345 **B)** ERR3486098 and ERR3764349 **C)** ERR3486099 and ERR3764351. The long and short read combinations analyzed from *Mus musculus* were **D)** ERR2680378 and ERR2680375 **E)** ERR2680378 and ERR2680377 **F)** ERR2680380 and ERR2680379. The long and short read combinations analyzed from human were **G)** SRR4235527 and NA12878-cDNA **H)** SRR4235527 and SRR4235527 **I)** SRR1153470 and SRR1163655.

over the long-read assemblies, and 4.1% over the short-read assemblies. The increase in the number of annotated transcripts was 21.7% and 5.0% over the long-read and short-read assemblies respectively. When comparing the results of hybrid-read assembly to an assembly of corrected long reads, the hybrid-read assembly had a very small decrease in precision of 1.0%, but an increase in the number of annotated transcripts of 14.4%. Finally, using the TALC-corrected long reads instead of the uncorrected long reads in a hybrid-read assembly only increased precision by 0.5% and increased the number of annotated transcripts by 0.4%.

### Mus musculus

In the *Mus musculus* samples, the hybrid-read assemblies showed an even greater improvement in precision versus the long-read only and short-read only assemblies (Fig 3D–3F). The percent increase was 38.6% over the long-read assemblies and 18.9% over the short-read assemblies. The number of annotated transcripts assembled increased substantially over the long-read assemblies with a relative increase of 118%; however, there was a slight decrease over the short-read assemblies of 0.6%. As with *Arabidopsis thaliana*, we saw that the hybrid-read assemblies outperform the corrected long-read assemblies with a 24.3% increase in precision and a 96.0% increase in the number of annotated transcripts. Hybrid-read assemblies using the TALC-corrected long reads again did not appear considerably different than the hybrid-read assemblies with the uncorrected reads: precision decreased by 0.5% while the number of annotated transcripts increased by 0.8%.

### Human

In the human data, we saw an increase in precision of 26.0% in the hybrid-read assemblies over the long-read assemblies, and an increase of 22.7% over the short-read assemblies (Fig 3G–3I). The number of annotated transcripts was also higher in the hybrid-read assemblies with an increase of 47.2% over the long-read assemblies and 36.5% over the short-read assemblies. As with the *Arabidopsis thaliana* and *Mus musculus* samples, the hybrid-read assemblies were still better than corrected long-read assemblies with 21.4% greater precision and 45.0% more annotated transcripts. The increase in precision and number of annotated transcripts in the hybrid-read assembly with corrected long reads compared to hybrid-read assembly with the uncorrected reads was again small, at 1.1% and 1.0% respectively. Because the human genome is the largest of the 3 genomes, we also compared the runtime of hybrid-read assembly to that of TALC. On average, hybrid-read assembly of the human samples took 48.8 minutes using 1 thread. In comparison, TALC took an average of 7143 minutes using 12 threads.

It is worth noting that long-read sequencing technology, including sequencing chemistry and basecalling, has improved since the long reads in this analysis were generated. We also ran a similar analysis on a direct RNA ONT dataset generated with newer chemistry (SQK-RNA002) and basecalled with Guppy v5.0.7 [13]. Just as with the older data, the hybrid-read assembly achieves better accuracy than either the long or the short-read assembly. We also used these data to evaluate the level of support of the assembled transcripts according to the RefSeq annotation. As expected, the hybrid-read assembly contains more curated (highly supported) transcripts and more predicted (poorly supported) transcripts. The results are shown in S2 File and S3 Fig.

### Annotation-Guided assembly

As with the simulated data, we also performed annotation-guided assembly for each dataset shown in Fig 3 and evaluated the precision (Fig 4A) and number of annotated transcripts assembled (Fig 4B). We compared these results to the hybrid-read assemblies created without

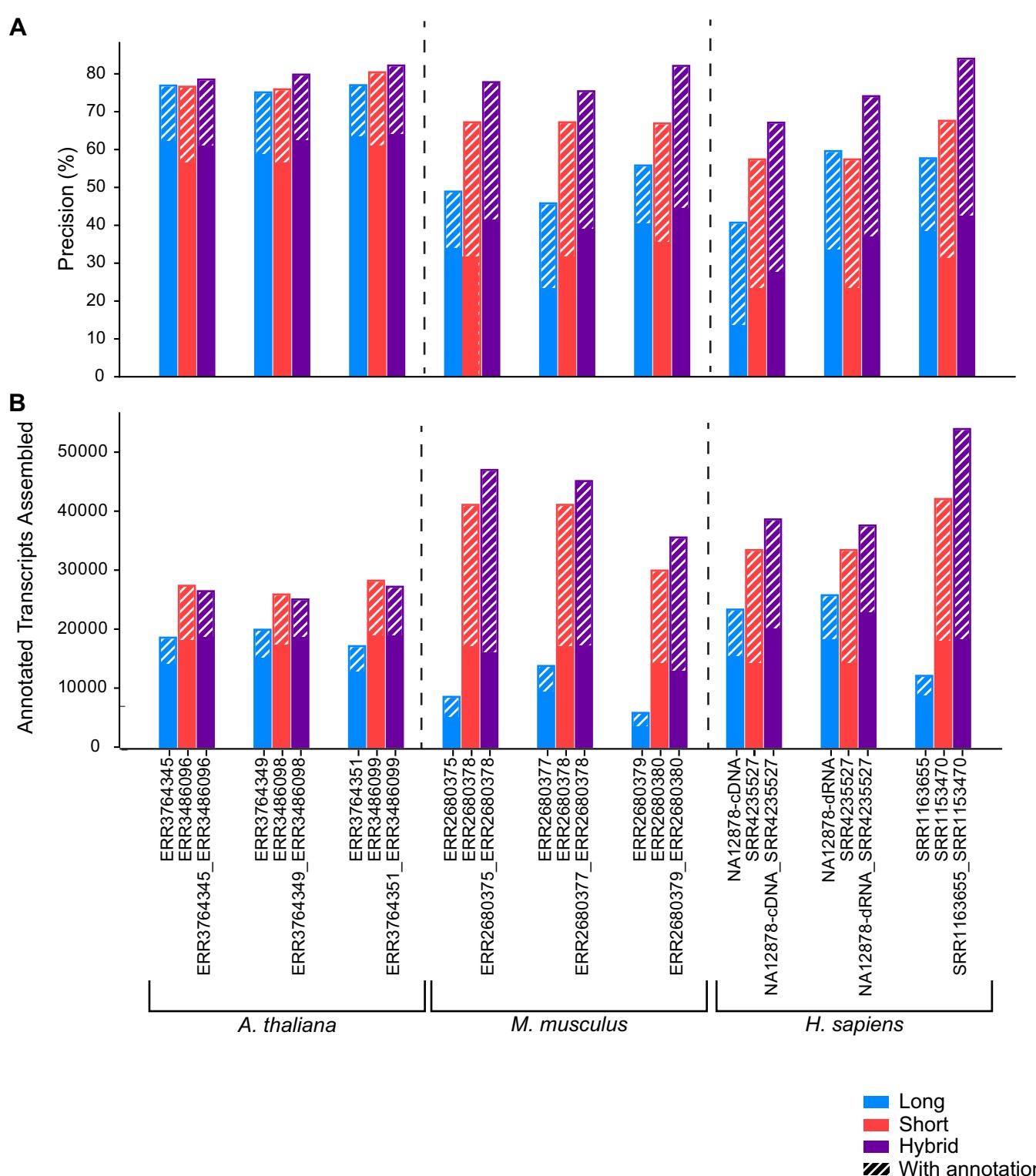

**Fig 4. A)** Precision of assemblies of all real datasets with and without guide annotation **B)** The number of annotated transcripts assembled in assemblies of all real datasets with and without guide annotation.

guide annotation. The average precision of the *Arabidopsis thaliana* hybrid-read assemblies increased from 62.5% to 80.2% and the average number of annotated transcripts assembled increased from 18,711 to 26,214. The average precision of the *Mus musculus* hybrid-read assemblies increased from 41.7% to 78.4% and the number of annotated transcripts increased from 15,342 to 42,541. Lastly the precision of the human assemblies increased from 35.7% to 75.1% and the number of annotated transcripts assembled more than doubled, increasing from 20,369 to 43,363. Across all samples in all species, the annotation-guided hybrid-read assemblies had greater precision than the annotation-guided long and short read assemblies. In all *Mus musculus* and human samples, the hybrid-read assemblies also contain a greater number of annotated transcripts. The *Arabidopsis thaliana* assemblies contain more annotated transcripts than the long-read assemblies, but slightly fewer than the short-read assemblies.

## Discussion

The new StringTie algorithm described here uses the strengths of both long- and short-read RNA-seq data to improve transcriptome assembly. By using the short reads to support or adjust splice sites identified in the long-read alignments, we were able to reduce noise caused by the high error rate of long reads. Using simulated data, we demonstrated that hybrid-read assemblies achieve greater precision and sensitivity than both the long-read only and short-read only assemblies across a range of sensitivity parameters. We also showed that the calculated transcript coverage correlates better with the true coverage in the hybrid-read assemblies. Using real data from 3 different species, we showed that hybrid-read assemblies are more precise than long and short-read assemblies across all samples in all species. The hybrid-read assemblies also contained more transcripts that precisely matched the reference annotations as compared to the long and short-read assemblies in all but 2 *Mus musculus* datasets (Fig 3D and 3F). In these 2 datasets, the hybrid-read assemblies contained more annotated transcripts than the long-read assemblies, but slightly fewer than the short-read assemblies.

Performing hybrid assembly with the new StringTie algorithm is akin to correcting the long reads prior to assembly; therefore, we compared StringTie's hybrid assembly to assembling long reads corrected by TALC. Notably, read correction with TALC took 146 times longer to run than StringTie on human data. Furthermore, we found that all of the hybrid-read assemblies contained more annotated transcripts than the assemblies of TALC-corrected long reads. All but 2 *Arabidopsis thaliana* hybrid-read assemblies also achieved greater precision. We also tested whether using corrected long reads in a hybrid-read assembly would be more accurate than using uncorrected reads. As shown in Fig 3, the difference between using corrected versus uncorrected long reads with StringTie's hybrid algorithm is very small, ranging from ~0.5% to 1% for both precision and the number of annotated transcripts assembled. When considering the substantial increase in runtime and the marginal increase in accuracy, we conclude that using StringTie's hybrid assembly algorithm with uncorrected long reads is the preferable method of transcriptome assembly. The lowest error rate observed in any of the long-read datasets after correction was 1.8% in the human PacBio data. Hybrid-read assembly was still more accurate than this corrected long-read assembly suggesting that even as long-read technology inevitably improves and error rates decline, hybrid-read assembly will still offer benefits over long-read only assembly.

Because *Arabidopsis thaliana*, *Mus musculus*, and human are well-studied organisms, they have high-quality reference annotations. This allowed us to perform separate experiments in which StringTie was run with a guide annotation. Across all datasets among the simulated and real data we saw substantial improvements in accuracy. This evidence indicates that the best results are achieved with annotation-guided hybrid assembly for species with high-quality

reference annotations. We have demonstrated that hybrid-read assembly with StringTie is better than long-read, short-read, or corrected long-read assemblies. As the first reference-based, hybrid-read transcriptome assembler, we believe this new release of StringTie will be a valuable tool leading to improvements in transcriptomic studies of many species.

## Methods

### StringTie algorithm for hybrid data

As previously described, StringTie takes as input an alignment file of all reads from a sample in either SAM or BAM format [14], uses these alignments to create a splice graph, and then assembles transcripts by iterating through two steps: first, it identifies the heaviest path in the splicing graph and makes that the candidate transcript; and second, it assigns a coverage level to that transcript by solving a maximum-flow problem [8]. If a known annotation is provided as input to StringTie, then the first step above is initially restricted to paths in the splicing graph that correspond to transcripts in the annotation. After all transcripts in the annotation have been exhausted, if there are still paths in the splicing graph that are covered by reads, the algorithm resumes using its default heuristic to identify the heaviest path in the graph. This new release of StringTie follows the same two steps to assemble transcripts, but also supports input alignment data in CRAM format as it now makes use of the HTSlib C library [15] and can operate in *hybrid data* mode, enabled by the `--mix` option. In this new mode of operation, StringTie takes as input two alignment files, the first file on the command line containing the short-read alignment data and the second one having the long-read alignments. These two alignment files are parsed in parallel to identify clusters of reads that represent potential gene loci. Errors in the reads or the alignments, which are commonly present in the long-read data, propagate to the construction of the splice graph, creating vastly more paths through the graph, which not only slows down the algorithm, but also makes it much more difficult to choose the correct set of isoforms (each of which corresponds to a path) at a particular gene locus. As illustrated in Fig 5, each mis-aligned long read can create a "noisy" transcript that appears to have alternative donor and acceptor sites, extra exons, or skipped exons. In the figure, we show two noisy transcripts, one with an extra exon and an erroneous acceptor (AG) site, and the other with two erroneous donor (GT) sites. These two noisy transcripts together contribute four additional exons to the splice graph, shown on the right side of the figure. These additional exons then generate 8 additional, erroneous edges in the graph, shown in

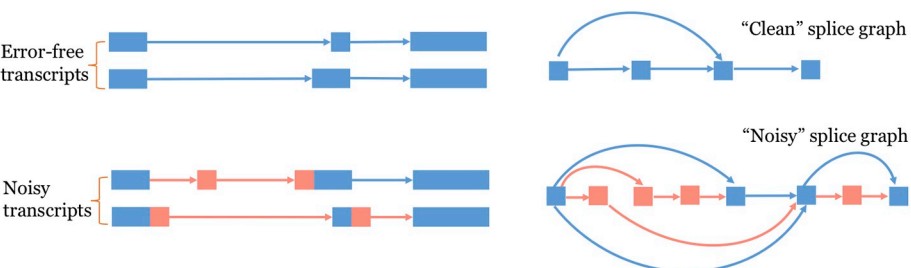

**Fig 5. Noisy alignments make the splice graph vastly more complicated.** The clean splice graph on the upper right is based on the two error-free transcripts, while the noisy splice graph is based on all four of the transcripts shown on the left. Regions shown in orange are errors due to mis-alignments.

orange. Thus, while the clean splice graph has only 4 nodes and 4 edges, the noisy splice graph has 8 nodes and 11 edges. Because every possible path through a splice graph is a possibly valid isoform, the number of isoforms grows exponentially as we add edges. In this simplified example, the clean splice graph shown on the upper right, based on 2 error-free transcripts, has only 2 paths, each representing a correct transcript. The noisy splice graph, in contrast, has 8 possible paths, only 2 of which correspond to genuine transcripts. Note that a splicing graph implicitly assumes independence of local events, and thus it typically contains many more legal paths than the number of transcripts used to create it.

With a hybrid data set containing both long and short reads, we can take advantage of highly accurate short reads to fix most of these problems. The strategy we employ is to scan all the splice sites at a locus in order to evaluate how well-supported each site is by the read alignments. If a splice site is not well-supported (e.g., by at least one short read, or by most of the long reads that have splice sites in a small window around that particular splice site), we will search for a nearby splice site with the best support (i.e. one that has the largest number of alignments agreeing with it), and adjust the long-read alignment correspondingly. We found that this strategy can greatly reduce the number of spurious splice sites. Relying on short read data, we can also fix other long read alignment artifacts. For instance, one common problem that we and others [16] noticed is the ambiguity of strand of origin for long reads. Due to their high error rate, the aligner sometimes infers the wrong strand for the long-read alignment. We can fix this by scanning nearby splice sites, and choose the strand of the alignment that is best supported by the short-read data. Another common problem is the presence of false "exons" introduced by insertions in the long reads. These insertions tend to be small (usually less than 35bp), so to address this issue, we remove exons that have support only from long reads and that are contained within introns that are well supported by short read alignments.

After the splice graph has been pruned to remove erroneous splice sites and nodes, the hybrid version of StringTie will execute the next two steps:

1. First, it will cluster all compatible long-read alignments. We can do this efficiently by taking advantage of the sparse bit vector representation of the splice graph already employed by StringTie, where each node or edge in the graph corresponds to a bit in the vector. A read or a paired read (in the case of short read data) will therefore be represented by a vector of bits where only the bits that represent the nodes or edges spanned by the read and its pair are set to 1. The bit representation provides a quick way to check compatibilities between long reads. Each cluster will represent a path in the splice graph that will have an initial expression level estimate E(l) based on the number of long reads covering that path. Note that a cluster does not always have to be a full transcript (i.e. if all long reads in the cluster come from a truncated cDNA molecule), although in most cases it will be.

2. For each cluster path P inferred in the previous step, starting from the one with the largest number of long reads supporting it, StringTie will use the short-read alignment to output an assembled transcript and expression level estimate. First, StringTie will choose the heaviest path in the splice graph that includes P. This will represent a candidate transcript. Then StringTie will use its maximum flow algorithm to compute an expression level estimate E(s) based on short-read data only. The final expression level of the transcript will be equal to E(l)+ E(s), and short-read alignments that contribute to E(s) will be removed from the subsequent expression level computations.

Note that some gene loci might have either only long-read or short-read alignments present. For those cases, StringTie will follow its previously implemented algorithms to assemble those loci [9].

### Reference genomes and annotations

The human reads (simulated and real) were aligned to GRCh38 and compared to the RefSeq annotation version GRCh38.p8 for accuracy. The *Mus musculus* reads were aligned to GRCm39 (GenBank Accession GCA_000001635.9) and accuracy was computed using the GENCODE annotation version M26. The *Arabidopsis thaliana* reads were aligned to TAIR10.1 (GenBank Accession GCA_000001735.2)

### Simulated data generation

We used the same short read simulated data from FluxSim [17] as was used to evaluate String-Tie2 [9] We used NanoSim [18] to simulate ONT direct RNA sequencing reads. Using the NA12878-dRNA reads, we built a model of the reads by using the read_analysis.py module of NanoSim in transcriptome mode with the following command:

```
read_analysis.py transcriptome -i ONT_dRNA_reads.fq -rg GRCh38.fa -rt
transcripts.fa -annot hg38c_protein_and_lncRNA_sorted.gtf -o training
```

where `transcripts.fa` is the human reference transcriptome obtained by using gffread [19].

We simulated 13,361,612 reads (the same number of reads in the NA12878-dRNA sample used build the model) by running the simulator.py module of NanoSim in transcriptome mode with the following command:

```
simulator.py transcriptome -rt transcripts.fa -rg GRCh38.fa -e expres-
sion_levels.tpm -r dRNA -n 13361612 -fastq -o simulated_dRNA -b guppy
-c training
```

To match the expression levels of the long reads to the short reads, we used the.pro file generated by FluxSim to calculate the TPM of each transcript. These values were given as input to the NanoSim simulation with the -e parameter.

### Equal coverage simulation

To control for coverage in the simulated data, we first calculated the coverage of each dataset simply by summing the lengths of every read and dividing by the sum of the lengths of the transcripts expressed. Doing this we found that the coverage of the short reads was 164.9, the coverage of the long reads was 195.7, and the coverage of the hybrid reads was 356.1. To increase the coverage of the short reads, we reran FluxSim and increased the number of simulated reads from 150,000,000 to 323,636,363, and this resulted in a coverage of 355.9. To increase the coverage of the long reads, we re-ran NanoSim and increased the number of simulated reads from 13,361,612 to 24,306,253. This resulted in a coverage of 342. Because the lengths of the long reads vary extensively, unlike the short reads, the increase in coverage is not always proportional to the increase in the number of reads. Nonetheless the coverage of this dataset is much higher than the original coverage of 195.7 and quite close to the target value of 356.1. For both new simulations, the transcripts were simulated at the same TPMs as in the original simulation.

### Alignment and assembly

All short reads were aligned with HISAT2 with default parameters [20] using the following command:

```
hisat2 -x hisat2_index -1 short_reads_R1.fastq -2 short_reads_R2.
fastq -S short_aligned.sam
```

Long reads were aligned with Minimap2 [21] using the default parameters for spliced alignment with the following command:

```
minimap2 -ax GRCh38.fa long_reads.fastq -o long_aligned.sam
```

Alignment files were sorted and converted to BAM format using samtools [14]. Transcriptome assembly and quantification was done with StringTie version 2.2.0. We used the following StringTie commands to assemble the input alignment file for each assembly type:

- For long-read data: `stringtie -L long_reads.bam`

- For short-read data: `stringtie short_reads.bam`

- For hybrid data: `stringtie --mix short_reads.bam long_reads.bam`

In the case of annotation-guided assembly, we added to all commands above the following option: `-G reference_annotation.gtf`

## Accuracy analysis

We define sensitivity as $TP/(TP + FN)$ and precision as $TP/(TP + FP)$ where TP (true positives) are correctly assembled transcripts, FP (false positives) are transcripts that are assembled but do not match the reference annotation, and FN (false negatives) are expressed transcripts that are missing from the assembly. We used gffcompare [19] to obtain these metrics in addition to the number of annotated transcripts assembled. All numbers reported are at the 'transcript' level (as opposed to the intron or base level accuracy also reported by gffcompare). The 'true positive' reference sets provided to gffcompare (with the -r option) are as follows:

Simulated data with varying sensitivity parameters (Fig 1A): Human reference transcripts fully-covered by either the long or short simulated reads. We define full coverage for multi-exon transcripts as coverage across all splice sites. For single-exon transcripts, it is considered fully covered if there is coverage across $> = 80\%$ of the length.

Simulated data with default sensitivity parameters (Fig 1C and 1D): The full set of expressed transcripts in the simulated data.

Real data (Figs 3 and 4): The reference annotation for the given species filtered to only include loci covered by at least one long read.

The -Q option was used with gffcompare to only consider loci present in the reference set provided.

Our main metric used to compare the accuracy of the long, short, and hybrid-read assemblies is relative percent increase in sensitivity and precision which is defined as $(S_1-S_2)/S_2$ and $(P_1-P_2)/P_2$ where $S_1$ and $P_1$ are the sensitivity and precision of the hybrid-read assembly and $S_2$ and $P_2$ are the sensitivity and precision of the assembly we are comparing it to. For example, a 10% absolute increase in sensitivity from $S_2 = 20\%$ to $S_1 = 30\%$ results in a relative increase of 50% [9]. For the real data, S is the number of annotated transcripts assembled.

## Coverage analysis of simulated data

The expected coverage for the long-read only and short-read only assemblies was obtained by taking the sum of the lengths of all the reads covering a transcript, and dividing it by the transcript length. For the hybrid-read assemblies, the expected coverage was calculated by taking the sum of the short-read and long-read expected coverages of each transcript. The computed read coverages were taken from StringTie's output for each type of assembly. All coverages were exported to R and normalized to $\log_2(1 + \text{coverage})$. To make the comparison fair, we

only plotted the coverages and calculated the $R^2$ for the transcripts that were shared between the long-read only, short-read only, and hybrid-read assemblies.

## Long-read correction with TALC

For each set of long reads, we first counted all 21-mers in the short reads from the sample using Jellyfish [22]. The kmer counts were obtained with the following commands:

```
jellyfish count -mer 21 -s 100M -o kmers.jf -t 8 $short_reads_1.fa
$short_reads_2.fa
jellyfish dump -c kmers.jf > kmers.dump
```

   Using the Jellyfish output, we ran TALC with the following command:

```
talc $long_reads.fa -SRCounts kmers.dump -k 21 -o $long_reads_TALC.fa
-t 12
```

## Error-rate calculations and full-length isoform analysis

Annotated transcript sequences for each species were extracted from the reference genome using gffread [19] with the following command:

```
gffread -w transcripts.fa -g reference_genome.fa reference_annota-
tion.gtf
```

   We then aligned each long-read dataset to the transcript sequences using Minimap2 and output the alignments in PAF format. To calculate the indel and mismatch rates, we first selected the primary alignment for each read. To calculate the indel rate, we summed the number of insertions and deletions in the alignment (using the CIGAR string) and divided by the alignment length (column 11 in the PAF output). To calculate the mismatch rate, we subtracted the number of matches (column 10 in the PAF output) and the number of insertions and deletions from the alignment length and divided the result by the alignment length. The total error rate is the sum of the indel and mismatch rates.

   To identify full-length isoforms, we filtered for reads that spanned all intron/exon boundaries of a multi-exon transcript or 80% of the length of a single-exon transcript. The reference annotations were used to identify the coordinates of the intron/exon boundaries.

## Supporting information

**S1 Fig. Transcript assembly accuracy at all expressed loci in short, long, and hybrid simulated data sets. A)** Sensitivity and precision of the assemblies created from the original dataset where the hybrid read coverage is the combination of the long read and the short read coverage. **B)** Sensitivity and precision of the assemblies created from the dataset where the coverage of the short, long, and hybrid reads is approximately equal. The two StringTie parameters varied were the minimum read coverage allowed for a transcript (-c) and the minimum isoform abundance as a fraction of the most abundant transcript at a given locus (-f). Each shape represents a different combination of -c,-f parameters with the values indicated in the legend. (EPS)

**S2 Fig. Sensitivity and the number of annotated transcripts assembled for 9 real datasets from *Arabidopsis thaliana*, *Mus musculus*, and human.** All loci are included in these calculations. The circle markers represent assemblies created from uncorrected reads, and the stars represent assemblies created from long-reads corrected with TALC. The long and short read combinations analyzed from Arabidopsis thaliana were **A)** ERR3486096 and ERR3764345 **B)** ERR3486098 and ERR3764349 **C)** ERR3486099 and ERR3764351. The long and short read combinations analyzed from Mus musculus were **D)** ERR2680378 and ERR2680375 **E)** ERR2680378 and ERR2680377 **F)** ERR2680380 and ERR2680379. The long and short read

combinations analyzed from human were **G)** SRR4235527 and NA12878-cDNA **H)** SRR4235527 and NA12878-dRNA **I)** SRR1153470 and SRR1163655.
(EPS)

**S3 Fig. Transcript assembly accuracy on RNA-seq data from the HepG2 cell line. A)** Precision and number of annotated transcripts assembled from long, short, and hybrid-read assemblies generated from reads from the HepG2 cell line. **B)** The number of predicted and curated transcripts assembled in the long, short, and hybrid-read assemblies generated from reads from the HepG2 cell line. Predicted means that the transcript is poorly supported according to the RefSeq annotation and curated means the transcript is highly supported.
(EPS)

**S1 Table.** Error rates of all long-read datasets before (a) and after correction (b) with TALC.
(DOCX)

**S2 Table. Percentage of reads that are full-length isoforms and the number of unique full-length isoforms captured in each long-read dataset.**
(DOCX)

**S1 File. Equal Coverage Simulation Results.**
(DOCX)

**S2 File. Hybrid transcriptome assembly of short and long read data from the HepG2 cell line.**
(DOCX)

## Acknowledgments

Authors want to thank Steven Salzberg for proofreading the manuscript.

## Author Contributions

**Conceptualization:** Mihaela Pertea.

**Data curation:** Brandon Wong, Geo Pertea.

**Formal analysis:** Alaina Shumate, Brandon Wong, Mihaela Pertea.

**Funding acquisition:** Mihaela Pertea.

**Methodology:** Mihaela Pertea.

**Software:** Geo Pertea, Mihaela Pertea.

**Supervision:** Mihaela Pertea.

**Validation:** Alaina Shumate, Geo Pertea, Mihaela Pertea.

**Visualization:** Alaina Shumate, Brandon Wong, Mihaela Pertea.

**Writing – original draft:** Alaina Shumate, Mihaela Pertea.

**Writing – review & editing:** Alaina Shumate, Geo Pertea, Mihaela Pertea.

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
