## [Decision Letter · Decision Letter 0]

26 Jan 2022

Dear Dr Pertea,

Thank you very much for submitting your manuscript "Improved Transcriptome Assembly Using a Hybrid of Long and Short Reads with StringTie" for consideration at PLOS Computational Biology.

As with all papers reviewed by the journal, your manuscript was reviewed by members of the editorial board and by several independent reviewers. In light of the reviews (below this email), we would like to invite the resubmission of a significantly-revised version that takes into account the reviewers' comments.

Both of the reviewers made very positive comments and liked this research topic, while they also made some critical suggestions for the author to revise the manuscript. Especially, reviewer 1 mentioned how to choose simulated data sets and real data sets to strengthen the results.

We cannot make any decision about publication until we have seen the revised manuscript and your response to the reviewers' comments. Your revised manuscript is also likely to be sent to reviewers for further evaluation.

Sincerely,

Jinyan Li

Associate Editor

PLOS Computational Biology

Ilya Ioshikhes

Deputy Editor

PLOS Computational Biology

Both of the reviewers made very positive comments and liked this research topic, while they also made some critical suggestions for the author to revise the manuscript. Especially, reviewer 1 mentioned how to choose simulated data sets and real data sets to strengthen the results.

Reviewer's Responses to Questions

**Comments to the Authors:**

Reviewer #1: Shumate et al describe a new release of the transcriptome assembler StringTie. In this manuscript, they set out to determine whether hybrid assemblies generated with Illumina and Nanopore data result in greater accuracy than assemblies generated with just one data type. In principle, the results for three higher eukaryotes support their claims although I think a deeper analysis of the data would prove more informative and strengthen the manuscript considerably.

Major comments

1. The Abstract describes long read sequencing as low throughput. While this is true when comparing a MinION to a high output Illumina platform (NextSeq, HiSeq, NovaSeq) but now when comparing against the GridION or PromethION.

2. Introduction.

a. The first is that proteomic diversity and phenotypic complexity is not exclusively limited to higher eukaryotes. Indeed many plants, bacteria, and viruses exhibit remarkably complex transcriptomes.

b. The second is that the error-rate quoted for nanopore direct RNA-Seq is way out of date. Indeed the most recent chemistries (SQK-RNA002) combined with later Guppy versions (v4 onwards) reduce the error rate to ~3-4%.

c. Finally, the authors state that practical limitations impede the ability to capture full-length transcripts. It would help significantly if the authors could put actual numbers on this. For instance, in our hands (and others), we typically observe between 30-60% full-length transcripts (SQK-RNA002 + Guppy v4).

3. Results

a. I have many questions about the underlying data, both simulated and real. In the case of the ONT data, it is imperative to include information on the Chemistry (SQK-RNA001 or SQK-RNA001 for DRS), basecalling (which Guppy versions) as this will very clearly impact on the accuracy of the final ONT fastq files, and the resulting alignments. Indeed one suspect for the latest chemistry + basecaller combinations that the numbers of artefacts in DRS datasets (false exons, incorrect splice junctions) will significantly reduce. These queries also extend to the simulated data. What do the simulated DRS datasets look like? Are they modelled on SQK-RNA001 or SQK-RNA002 chemistry? Similar questions pertain to the Illumina data. For instance, was this all poly(A) selected or were these ribo-depleted libraries?

b. Extending this further, do the authors see any difference in transcriptome assembly quality when comparing chemistry or basecaller versions? While it is abundantly clear that hybrid assembly is still the best option, it would be valuable to know whether further improvements to chemistry/basecalling would increase the relative accuracy of DRS alone approaches.

c. For the real world data, it would great if table 1 were to include more information on the source/cell types sequenced and also how the various datasets link together. For instance, Fig 3 pairs together Nanopore and Illumina datasets but it is now clear how/why these pairs specifically were chosen.

d. A deeper dive into the assembled transcriptomes is warranted. For instance, transcripts in the existing HG38 annotation are variously classified according to support (experimental, theoretical, etc). It would be interesting to know, for a given hybrid assembly, how many poorly supported transcripts in the existing annotation are actually confirmed by hybrid assembly?

e. The comparison to TALC is useful but what about other assemblers that offer ‘hybrid-like’ assembly? For instance, FLAIR uses Illumina data to infer splice junctions that are then ‘corrected’ in nanopore-derived alignments. While I suspect the StringTie approach to be faster and more accurate, it would be useful to demonstrate this definitively.

4. Methods

a. Linking back to comment 3a, more information on how NanoSim was run and what the resulting data look like is needed. What % of NanoSim reads are full length and how does this compare to the real-world datasets? What chemistry (SQK-RNA001 or SQK-RNA002) and basecalling (Guppy version) is simulated?

Minor comments:

1. Line numbers and page numbers would have made this manuscript easier to review! I do however appreciate the inclusion of figures within the manuscript body rather than at the end as this makes it much easier to read.

2. While I understand the choice of red, blue, purple for the figures, these colours are not easy to differentiate. The authors could consider lightening the shade of each or finding an alternative combination that segregates better.

3. It would be useful to have the test datasets available (or at least linked to) within the repository. Alternatively, a collection of ‘light’ test datasets containing reads for a just a few select transcripts would be useful for those who which to explore the various parameters and outputs.

4. In the Methods section, it would be useful to spell out the commands used for HiSat2 and MiniMap.

5. Was any filtering performed on the RefSeq annotations?

Reviewer #2: The manuscript " Improved Transcriptome Assembly Using a Hybrid of Long and Short Reads with StringTie" by Schumate et al. describes the authors' latest developments on the StringTie software, incorporating the capability to assemble transcript alignments from both short (tens to hundred base length Illumina reads) and long (several kilobase length PacBio or ONT) RNA-seq reads. The initial version of StringTie (2015, Nature Biotechnology) assembled short RNA-seq read alignments, StringTie2 (2019, Genome Biology) had specializations for assembling long read alignments, and now the latest version of StringTie (perhaps more aptly named StringTie3?) is presented here for co-assembly of short and long read alignments. The authors demonstrate with both simulated and real RNA-seq data that this latest StringTie provides more accurate assemblies when both the short and long read alignments are available.

StringTie continues to be one of the most popular genome-guided alignment assemblers for reconstructing transcripts, and is my personal favorite for routine use in my own work. I'm pleased to see that this new functionality of assembling both short and long reads is now integrated.

The manuscript is very well written and the experiments performed are all relevant and necessary for this work. I do have a few critical comments and suggestions as follows:

Major

When combining short read alignments with the long read alignments, the effective coverage of transcripts will increase. Within low to moderately expressed transcripts, increased effective coverage would be expected to yield increased rates of full reconstruction, separately from read types provided. The authors might provide some supplementary study to demonstrate that the improved accuracy for reconstruction with hybrid data is more due to having the hybrid data types available rather than due to increased effective alignment coverage. This could be done with simulated data by controlling for total effective coverage when assembling in short-only, long-only, or hybrid data.

Figure 1 would ideally show a more impactful example of where both read types are needed to assemble the full-length isoform. In the example shown, a splicing graph based on short read alignments alone looks as though it would contain the full-length splicing pattern, and if it was not assembled by StringTie, it might be due to StringTie parameter settings or StringTie-specific logic. A different, even naive splice graph assembler has the potential to reconstruct the full-length splicing pattern based on just those short reads alone.

It is useful that StringTie can make use of 'dirty' long reads (those with high error rates), but it is worth noting that the modern PacBio IsoSeq and ONT transcriptome sequencing methods are currently producing much higher fidelity data. The authors might comment on this in the manuscript. Characteristics of the real long read data being used with StringTie in this manuscript should be made available, perhaps as supplementary materials, to shed insight into the overall quality of those data - ie. are they early error-prone or later higher fidelity long reads being leveraged. In particular, it would be helpful to know characteristics (ie. percent identity or error rates) before and after applying TALC for error correction.

In Figures 2 and 3 that provide accuracy statistics for the different StringTie invocations, it would be helpful to know for the long reads what the full-length sensitivity and specificity would be without doing any StringTie assembly (for example, running a method like cd-hit to reduce redundancy, and examine accuracy statistics for the cd-hit compiled reference isoforms aligned with minimap2). This would provide a nice reference point for defining the baseline quantity of full-length isoforms from which StringTie would further augment through assembly.

The experiments performed with real data involved gathering transcriptome data from public databases. Unlike the simulated data, where the long and short reads appear to be derived from matched data (same genes being expressed), it isn't clear that the real data are similarly matched (so potentially many different genes being expressed between long and short read data). Table 1 doesn't indicate the tissue type that the data correspond to, and I couldn't easily ascertain it from the original source either. Accuracy analysis leveraging these real data were restricted to those genes having long read coverage, although the authors state (indicating data not shown) that results are similar when all loci are examined. There are a couple of issues to comment on here. First, I encourage the authors to provide (in supplement) results observed when considering all loci. This will address the issue of how much more comprehensive and accurate an assembly derived from StringTie when provided all data as opposed to just short or long reads alone, which would be useful to know in practice. Second, by restricting to only loci with long reads, because the derived samples may be unmatched, there is little assurance that relevant genes expressed with long reads will be expressed in the samples from which the short reads were derived. This disparity might account for lower sensitivity of short-only as compared to long-only in Figures 3G,H. A paucity of short reads for a long-read-covered locus would also presumably impact the ability of TALC for error correction at corresponding loci, but I suspect that has negligible effect on StringTie reconstruction given TALC-related results from other experiments presented here.

Figures 2C,D and Figure 4 involve barplots showing sensitivity and specificity separately instead of plotting together in scatter form (as in Figure 3). I think it's more transparent to assess sensitivity and specificity together in scatter form rather than separately in the barplots, unless there's some specific reason to show them separately here.

Minor

Abstract: "unable to span multiple exons" -> "rarely span multiple exons" (as 'unable' is not true, and 'rarely' is used elsewhere in the manuscript).

Intro: " Additionally, long reads have a high error rate between 13% and 15% [4], and the throughput of long-read RNAseq is much lower than that of short-read RNA-seq." This was a true statement for the earlier state of the technology, but due to recent rapid advances, this is no longer true. This recent advance would be worthy of commenting on in the Intro.

Reference annotation-assisted assembly has been used by StringTie and earlier methods to leverage reference isoform structures as part of generating a more complete assembly. I encourage the authors to include more information in the methods on how StringTie does this, or reference earlier StringTie paper(s) that describe it sufficiently. In particular, describing how it contributes to improving specificity of reconstruction would be most helpful, as I find it less intuitive than improving sensitivity.

Please review figure 5 to ensure that the "Noisy" splice graph accurately reflects the read alignments. I think the bottom pink arrow needs to be shifted to the right.

Please be sure to make inputs (aligned bams, simulated fastqs, etc.) available on a data sharing site.

**Have the authors made all data and (if applicable) computational code underlying the findings in their manuscript fully available?**

Reviewer #1: Yes

Reviewer #2: **No: **I request the simulated data (fastqs) and aligned bams be made available on a data sharing site.

PLOS authors have the option to publish the peer review history of their article (what does this mean?). If published, this will include your full peer review and any attached files.

Reviewer #1: No

Reviewer #2: No
---

## [Decision Letter · Decision Letter 1]

14 Apr 2022

Dear Dr Pertea,

Thank you very much for submitting your manuscript "Improved Transcriptome Assembly Using a Hybrid of Long and Short Reads with StringTie" for consideration at PLOS Computational Biology. As with all papers reviewed by the journal, your manuscript was reviewed by members of the editorial board and by several independent reviewers. The reviewers appreciated the attention to an important topic. Based on the reviews, we are likely to accept this manuscript for publication, providing that you modify the manuscript according to the review recommendations.

In particular, Reviewer 2 made some technical comments about performance change when including vs. excluding the reference annotations; the reviewer also requested some clarification and further understanding about Figures 2 and 3. Another round of revision would make the manuscript more improved.

Sincerely,

Jinyan Li

Associate Editor

PLOS Computational Biology

Ilya Ioshikhes

Deputy Editor

PLOS Computational Biology

[LINK]

In particular, Reviewer 2 made some technical comments about performance change when including vs. excluding the reference annotations; the reviewer also requested some clarification and further understanding about Figures 2 and 3. Another round of revision would make the manuscript more improved.

Reviewer's Responses to Questions

**Comments to the Authors:**

Reviewer #1: I am quite satisfied with the detailed and considerate responses provided by authors. I thus recommend this paper be accepted for publication.

Reviewer #2: I appreciate the authors responses to reviewers comments and the manuscript is substantially improved and strengthened as a result of the additional experiments performed and related revisions. I have only a few more technical comments that the authors may further consider.

The results provided in Figure 4 indicate the accuracy differences between methods while leveraging the reference annotation as a guide. It would be useful to see the difference in accuracy for each sample as a result of including vs. excluding the reference annotations as well, as shown for simulated data in Figure 2C,D, providing those results in the supplementary materials if they are not easily integrated into the main figure.

In Figure 2C, it is unusual or unexpected that the precision appears largely unchanged for short reads when annotations are added as a guide. This would be worth reviewing and understanding further.

The results statement "The use of hybrid-read data plus annotation had an increase in precision of 10.7% and an increase in sensitivity of 23.5% as compared to using short reads plus annotation, which in turn was superior to using long reads plus annotation." on page 9 is not clear from Fig 2 with respect to how the short and long read (non-hybrid executions) rank compared to each other.

In Figure 3 and particularly in the new Supplementary Figure 2, the heavily increased precision for corrected long reads is strikingly different for Arabidopsis as compared to human and mouse. Similarly, the lower precision for hybrid model compared to corrected long reads is surprising and deserves explanation if possible. One feature of the Arabidopsis genome compared to mouse and human would be the compact gene structures and prevalence of retained introns in transcriptome sequencing data. Could this be partly responsible for the different behavior of Stringtie on Arabidopsis? Other reasons?

I appreciate that the authors added the read error characteristics in Table S1. This is very useful. However, this is limited to mismatches and doesn't include statistics on indels - which are most relevant to the long read data. Please include the indel rates as well, along with methods info on how this was done according to current best practices.

Minor

Figure 2B: should indicate in the legend that coverage refers to log2(1+coverage) as described in the Methods.

A small suggestion for wording: Page 9 "Although we cannot know exactly which transcript 'molecules' are present in the samples"

**Have the authors made all data and (if applicable) computational code underlying the findings in their manuscript fully available?**

Reviewer #1: Yes

Reviewer #2: Yes

PLOS authors have the option to publish the peer review history of their article (what does this mean?). If published, this will include your full peer review and any attached files.

Reviewer #1: No

Reviewer #2: No

Figure Files:

Data Requirements:

Reproducibility:

References:

---

## [Decision Letter · Decision Letter 2]

11 May 2022

Dear Dr Pertea,

We are pleased to inform you that your manuscript 'Improved Transcriptome Assembly Using a Hybrid of Long and Short Reads with StringTie' has been provisionally accepted for publication in PLOS Computational Biology.

Best regards,

Jinyan Li

Associate Editor

PLOS Computational Biology

Ilya Ioshikhes

Deputy Editor

PLOS Computational Biology

In the revised versions of your manuscript, you have addressed all of the concerns and suggestions from the reviewers. The reviewers are now satisfactory with these changes and revisions to the manuscript. I therefore recommend acceptance for the paper to be published.

Reviewer's Responses to Questions

**Comments to the Authors:**

Reviewer #2: Thank you for addressing my earlier concerns. Congratulations on a beautiful manuscript and your outstanding work on Stringtie.

**Have the authors made all data and (if applicable) computational code underlying the findings in their manuscript fully available?**

Reviewer #2: Yes

PLOS authors have the option to publish the peer review history of their article (what does this mean?). If published, this will include your full peer review and any attached files.

Reviewer #2: No

---

## [Editor Report · Acceptance letter]

27 May 2022

PCOMPBIOL-D-21-02222R2 

Improved Transcriptome Assembly Using a Hybrid of Long and Short Reads with StringTie

Dear Dr Pertea,

I am pleased to inform you that your manuscript has been formally accepted for publication in PLOS Computational Biology. Your manuscript is now with our production department and you will be notified of the publication date in due course.

With kind regards,

Livia Horvath
